# Understanding the Effects of Lactose Hydrolysis Modeling on the Main Oligosaccharides in Goat Milk Whey Permeate

**DOI:** 10.3390/molecules24183294

**Published:** 2019-09-10

**Authors:** Caroline Thum, Valerie Weinborn, Daniela Barile, Warren C McNabb, Nicole C Roy, Juliana Maria Leite Nobrega de Moura Bell

**Affiliations:** 1Food Nutrition & Health Team, AgResearch, 4442 Palmerston North, New Zealand; 2Riddet Institute, Massey University, 4442 Palmerston North, New Zealand; 3Department of Food Science and Technology, University of California, Davis, CA 95616, USA; 4High-Value Nutrition National Science Challenge, New Zealand; 5Department of Biological and Agricultural Engineering, University of California, Davis, CA 95616, USA

**Keywords:** lactose hydrolysis, goat milk oligosaccharides, β-galactosidase, transgalactosylation, whey, mass spectrometry, processing

## Abstract

Enzymatic hydrolysis of lactose is a crucial step to improve the efficiency and selectivity of membrane-based separations toward the recovery of milk oligosaccharides free from simple sugars. Response surface methodology was used to investigate the effects temperature (25.9 to 54.1 °C) and amount of enzyme (0.17 to 0.32% *w/w*) at 1, 2, and 4 h of reaction on the efficiency of lactose hydrolysis by *Aspergillus oryzae* β-galactosidase, preservation of major goat whey oligosaccharides, and on the de-novo formation of oligosaccharides. Lactose hydrolysis above 99% was achieved at 1, 2, and 4 h, not being significantly affected by temperature and amount of enzyme within the tested conditions. Formation of 4 Hexose (Hex) and 4 Hex 1 Hex and an increased de-novo formation of 2 Hex 1 N-Acetyl-Neuraminic Acid (NeuAc) and 2 Hex 1 N-Glycolylneuraminic acid (NeuGc) was observed in all treatments. Overall, processing conditions using temperatures ≤40 °C and enzyme concentration ≤0.25% resulted in higher preservation/formation of goat whey oligosaccharides.

## 1. Introduction

Milk oligosaccharides are sugars found in all mammalian species and possess diverse biological functions. These sugars are not digested by the suckling infant, remaining intact until they reach the large intestine where they are fermented and stimulate the growth of health-promoting bacteria. Oligosaccharides are highly concentrated in human milk (5–23 g·L^−1^) and are structurally diverse with more than 200 different oligosaccharides putatively identified [1]. Consumption of human milk oligosaccharides is also known to improve immune function [2], prevent adhesion of pathogens to intestinal epithelial tissues [3], increase absorption of minerals [4], and improve glucose homeostasis [5]. Functional studies, however, are limited due to the restricted supply of human milk as a source of oligosaccharides for large-scale isolation.

Alternative sources of human milk-like oligosaccharides have been identified in domestic animals such as bovine and caprine milk. The levels of oligosaccharides in the milk of domestic mammals, however, are much lower than the ones in human milk, typically less than 1.0 g·L^−1^ [6], which demands the development and application of enrichment methods to purify these biologically active sugars. Similarly to other ruminant species, caprine milk contains neutral oligosaccharides composed by various building blocks including Hex (Hexose, which can be glucose or galactose), HexNAc (which can be either N-Acetyl-Glucosamine or N-Acetyl-Galactosamine) whereas acidic oligosaccharides contain additional monosaccharides such as sialic acid, which can be found in the form of NeuAc (N-Acetyl-Neuraminic Acid) or NeuGC (N-Glycolylneuraminic acid). Previously studies reported the purification of caprine milk oligosaccharides from whey using a combination of ultrafiltration and size exclusion chromatography [7] or lactose hydrolysis and solid-phase extraction [8]. These methods, however, were suitable for small-scale purification [7] and were unable to completely remove lactose as a main contaminant of the oligosaccharide purified fraction [8].

Our group developed a novel approach to recover oligosaccharides from colostrum bovine whey permeate in high purity at pilot-scale [9]. The method relies on the integration of optimized processing conditions that favor maximum lactose hydrolysis and monosaccharide fermentation prior to oligosaccharides concentration by selective membrane filtration. This processing strategy has been recently applied for the recovery of caprine milk oligosaccharide (CMO) at pilot scale [10]. Recovery yields, however, were lower for goat milk (75%) compared to the ones obtained for bovine milk oligosaccharides (BMO) (95%). The specificity and efficiency of the enzyme β-galactosidase from *Aspergillus oryzae*, used in both methods above described, was previously shown to vary with reaction conditions (initial lactose concentration, temperature, enzyme to substrate ratio, and pH) [11,12,13]. Optimization of enzymatic hydrolysis should, therefore, be optimized for each specific substrate. Moreover, β-galactosidase from *A. oryzae* has a natural transgalactosylation activity which, under specific conditions, synthesizes galacto-oligosaccharides from the monosaccharides, glucose and galactose [14,15]. The effects of different reaction conditions on lactase transgalactosylation activity of β-galactosidase from *A. oryzae* in goat milk oligosaccharides have not yet been fully explored.

Because enzyme specificity might lead to significant changes in the oligosaccharide profile, including the degradation of naturally occurring oligosaccharides as well as the synthesis of new oligosaccharides, the present study was undertaken to evaluate the effects of processing conditions on the specificity of β-galactosidase from *A. oryzae* in relation to the CMO profile. Our hypothesis was that while complete lactose hydrolysis could be achieved in a wide range of processing conditions, some CMO could be degraded by the action of β-galactosidase from *A. oryzae* under certain processing conditions. In that view, the optimization of processing conditions leading to complete lactose hydrolysis and maximum preservation of CMO becomes a key step to improve recovery yields at pilot-scale. In addition, the possibility of producing new oligosaccharides under ideal reaction conditions (complete lactose hydrolysis and target CMO preservation), could lead to selective enrichment of prebiotic CMO, thus helping to improve process economics and feasibility.

The overall objective of the present research was to maximize the efficiency of lactose hydrolysis by *A. oryzae* β-galactosidase while achieving maximum preservation of biologically important CMO naturally present in caprine whey permeate. In addition, we also investigated the de-novo formation of oligosaccharides potentially operated by the transgalactosylation activity of β-galactosidase [16].

## 2. Results and Discussion

### 2.1. Lactose Hydrolysis Optimization by β-Galactosidase

The individual effects, as well as the simultaneous interaction of independent variables on lactose hydrolysis efficiency, were evaluated using a central composite design. The effects of temperature and amount of enzyme (%), at different reaction times, on lactose hydrolysis efficiency are shown in Figure 1. The hydrolysis of lactose by β-galactosidase activity was not significantly affected by temperature and amount of enzyme within the tested conditions. Overall, lactose hydrolysis above 99.4% was achieved at 1, 2, and 4 h in all experimental conditions evaluated. Average lactose hydrolysis of 99.5, 99.6, and 99.8% were achieved at 1, 2, and 4 h, respectively. These results are in agreement with previously reported data for the hydrolysis of lactose in bovine whey permeate, for which nearly 100% of lactose hydrolysis was obtained at pH 4.5, temperature from 40 to 50 °C, and amount of enzyme ranging from 0.1 to 0.25% [10,12]. Rodriguez-Colinas et al., for example, showed that 0.1% (*v/v*) β-galactosidases from *A. oryzae,* at 40 °C, pH 6.7 was able to hydrolyze up to 96% lactose in 1.5 h, producing approximately 7.0 g/L of GOS in bovine whey [15]. Others [11,17] also reported the activity of β-galactosidases from *A. oryzae*, however, comparison of the results is challenging because of the differences in substrate composition, reaction mode (batch or continuous), mode of enzyme use (free or immobilized), or purpose of the work.

### 2.2. Effect of β-Galactosidase on Caprine Milk Oligosaccharides

The effects of β-galactosidases from *A. oryzae* on total and individual oligosaccharides were specific to each experimental condition (Figure 2A) and reaction time evaluated. The regression model (Table 1) and contour curve (Figure 2B–D) shows that the total oligosaccharides abundance at 1, 2, and 4 h of reaction is influenced by the reaction temperature (X_1_) but not enzyme concentration (X_2_). The predictive model was statistically significant (F_cal_ > F_tab_) at *p* < 0.05 and was able to explain 82, 58, and 37% of the variation between the predicted and observed values at 1, 2, and 4 h, respectively. According to the predictive model (Table 1, Figure 2B–D), the total abundance of oligosaccharides decreases as reaction temperature moves away from the axial points (25.9 and 54.1 °C) towards the central point (40 °C). Within the range evaluated, the maximum and minimum abundance of oligosaccharides were observed at 25.9 and ~43 °C, respectively. The minimum abundance of oligosaccharides at ~43 °C is likely the result of increased lactase activity and/or stability when compared with 25.9 and 54.1 °C. Although this enzyme possesses activity from 30 to 65 °C, its optimum activity is around 50 °C (as described by the manufacturer).

In order to gain a better understanding of the changes happening at the individual oligosaccharide level, the effects of each experimental condition on the abundance of total oligosaccharides (Table 2), acidic oligosaccharides (Table 3), and neutral oligosaccharides (Table 4) were expressed as percentage decrease or increase compared to the control (untreated sample). Although an increase in the abundance of acidic oligosaccharides (7 to 189%) was observed in all experimental conditions evaluated when reaction time increased from 1 h to 2 h and 4 h (Table 2), the reduction in neutral oligosaccharides (−5% to −99%), observed in treatments with temperature near to the central point (30 to 50 °C), led to an overall decrease in total oligosaccharide abundance. Our results highlight that the transgalactosylation activity of the lactase used in this study was dependent not only on the temperature but it was also oligosaccharide specific.

Increased abundance of acidic oligosaccharides was due to an increase in the abundance of 2 Hex 1 NeuAc (6-Sialyl-lactose, MW 634.2189) (1 to 274% compared to control) and 2 Hex 1 NeuGC (N-glycolylneuraminyl-hexosyl-lactose, MW 650.2139) (6 to 249% compared to control) (Table 3). According to the regression equation (Table 1), the abundance of 6-Sialyl-lactose (2 Hex 1 NeuAc) and N-glycolylneuraminyl-hexosyl-lactose (2 Hex 1 NeuGc), at 1 and 2 h of reaction, was dependent on the reaction temperature but not on enzyme concentration. The predictive model, although statistically significant (F_cal_ > F_tab_) at *p* < 0.05, was able to explain the following variations between predicted and observed values at 1 and 2 h, respectively: 45 and 68% for 6-Sialyl-lactose and 52 and 50% for N-glycolylneuraminyl-hexosyl-lactose. According to the predictive model (Table 1) and Appendix A, the total abundance of these two oligosaccharides decrease as reaction temperature moves away from the axial points (25.9 and 54.1 °C) towards the central point (40 °C).

Other acidic oligosaccharides, 1 Hex 1 HexNAc 1 NeuAc (MW 675.2455), 3 Hex 1 NeuAc (796.2718), 2 Hex 1 HexNAc 1 NeuAc (MW 837.2983), 3 Hex 1 NeuGc (MW 812.2667), were highly hydrolyzed by the enzyme in most treatments and reaction times (Table 3). It is worth mentioning that the decreased abundance of acidic oligosaccharides such as 3 Hex 1 NeuAc (796.271) and 3 Hex 1 NeuGc (MW 812.2667) is consistent with the increased abundance of oligosaccharides such as 2 Hex 1 NeuAc (634.2189) and 2 Hex 1 NeuGc (650.2139). According to the regression equations (Table 1), changes in abundance of 1 Hex 1 HexNAc 1 NeuAc (MW 675.2455) at 4 h (*p* < 0.05 and R^2^ = 51%), 3 Hex 1 NeuAc (796.2718) at 1 h (R^2^ = 56%) and 2 Hex 1 HexNAc 1 NeuAc (MW 837.2983) at 1 and 4 h (R^2^ = 74% and R^2^ = 56%, respectively) of reaction time is influenced by the reaction temperature but not enzyme concentration within the range evaluated. Maximum abundance of these oligosaccharides can be obtained at temperatures near to the lower axial point (25.9 °C) (Appendix A). Treatments that increase concentrations of sialyl-oligosaccharides (i.e., treatment 1, 3, 10, 11, Table 3) may benefit oligosaccharides pilot-scale purification from whey, as these sugars have been linked to many beneficial health effects [18].

As described in the regression equations in Table 1 and contour curves (Appendix A), the abundance of the neutral oligosaccharides 3 Hex (MW 505.1764) (1, 2, and 4 h), 3 Hex 1 HexNAc (MW 708.2558) (1, 2, and 4 h), 2 Hex 2 HexNAc (MW749.2823) (1 h, 4 h), and 2 Hex 1 HexNAc (MW 546.2029) (1 h) were influenced by temperature, and only 2 Hex 1 HexNAc (4 h) was influenced by a combination of temperature and enzyme concentration. In general, predictive models for the abundance of most neutral oligosaccharides showed that increased abundance could be obtained at low temperatures (≤40 °C) or a combination of low temperatures and low enzyme concentration (≤0.18%). As observed in Table 2, increase or lower reduction of total neutral oligosaccharides was only observed in treatments with low temperatures (≤40 °C), i.e., treatment 1 at 1 h (92%) and 4 h (−5%) and treatment 3 at 1 h (−14%). The increase in total neutral oligosaccharides was due to the formation of two new structures, 4 Hex, in treatments with low temperature (≤40 °C), and 4 Hex 1 HexNAc, in most treatments (Figure 3). Predictive models for the synthesis of 4 Hex for 1, 2, and 4 h were able to explain from 74 to 95% of the variation observed between predicted and observed values and the regression equation was statistically significant (F_cal_ > F_tab_) (Table 1). The oligosaccharides 4 Hex and 4 Hex 1 HexNAc were not present in the control whey sample, demonstrating the ability of β-galactosidases from *A. oryzae* to generate de-novo neutral oligosaccharides using pre-existing structures as building blocks, while simultaneously hydrolyzing lactose into β-d-galactose and β-d-glucose (Table 2).

This study demonstrates, for the first time, the individual and combined effects of temperature and the concentration of β-galactosidase from *A. oryzae* on caprine milk oligosaccharides during lactose hydrolysis. We demonstrated that, although lactose hydrolysis above 99% was achieved at 1, 2, and 4 h of reaction, total oligosaccharide abundance varied with processing conditions evaluated. Overall, the use of low temperature (~25 °C) was shown to increase the abundance of total oligosaccharides, with higher abundance observed at a longer reaction time (4 h). Observed changes in the total abundance of oligosaccharides were related to reductions in the abundance of neutral oligosaccharides, the formation of two neutral oligosaccharides (4 Hex and 4 Hex 1 HexNAc) and increased formation of acidic oligosaccharides such as 6-Sialyl-lactose and N-glycolylneuraminyl-hexosyl-lactose observed for most processing conditions. In conclusion, processing conditions using lower temperatures (≤40 °C) and a lower amount of enzyme (≤0.25%) would lead to ideal results in terms of nearly complete lactose hydrolysis (>99%) and preservation/formation of neutral and acidic oligosaccharides, which may improve recovery yields of goat milk oligosaccharides during pilot-scale purification.

## 3. Materials and Methods

### 3.1. Goat Milk Source and Whey Production

Fresh raw goat milk was kindly provided by the UC Davis Goat Farm, Davis, CA, USA, from late lactation (8 months from kidding). Milk was initially defatted by the use of a cream separator (GEA Westfalia Separator, Model CTC 3, Oelde, Germany) and decaseinated by rennet Chy-Max^®^ Extra (100% chymosin, Chr. Hansen, Milwaukee, WI, USA) addition as previously described [10]. The whey obtained after the removal of caseins was frozen until subsequent lactose hydrolysis. Whey lactose concentration (see Section 3.4 for methods) and pH (Accumet AB15, Fisher Scientific, Waltham, Massachusetts, USA) were 51.78 g/L and 6.15, respectively.

### 3.2. β-Galactosidase Treatment of Goat Whey

Food-grade fungal lactase (Bio-Cat Inc., Troy, VA, USA) derived from the fungus *A. oryzae* (100,000 ALU/g)*,* with activity at pH ranging from 2.5 to 7.0 and temperature from 30 to 65 °C, was used to hydrolyze lactose into β-d-galactose and α-d-glucose. Goat whey was adjusted to pH 4.5 with 0.1 N HCl, and β-Galactosidase was added to achieve 0.17 to 0.32% (*wt/wt*) (Table 1). Approximately 13 g of whey permeate were incubated at various temperatures ranging from 25.9 to 54.1 °C (Table 1) for 1, 2, or 4 h at 50 rpm in 250 mL Erlenmeyer flasks. After each experimental condition was evaluated, samples were immediately centrifuged at 3900× *g* for 30 min using a 30-kDa molecular weight cut-off centrifugal filter device (Amicon Ultra-15 Centrifugal Filter, Millipore, Billerica, MA, USA) to separate the enzyme from the hydrolyzed fraction. Samples from all experiments were analyzed for glucose, galactose, lactose, and oligosaccharides profile.

### 3.3. Experimental Design and Statistical Analysis

Response surface methodology was used to investigate the optimal reaction parameters affecting the efficiency of lactose hydrolysis by β-galactosidase and the preservation of the major oligosaccharides present in goat whey. The use of a factorial design methodology, associated with response surface analysis, enables simultaneous analysis of multiple variables, thus reducing processing time and costs. The individual and combined effects of the most important parameters that affect β-galactosidase activity (temperature and amount of enzyme) were evaluated by a central composite rotatable design, with 3 central points and 4 axial points [19,20]. The total number of experiments followed the equation 2^k^ + 2k + nc, where k is the number of independent variables and nc is the number of repetitions in the central point. The effects of temperature *A. oryzae* (25.9 to 54.1 °C) and amount of enzyme (0.17 to 0.32% *wt/wt*) on the efficiency of lactose hydrolysis by β-galactosidase and preservation of major oligosaccharides were evaluated. The independent variables (temperature and amount of enzyme) were evaluated according to coded levels (−1.41, 0, +1, +1.41; Appendix A). The variable levels used in the experimental design were selected based on published data [10,12]. Previous data [10,12] also established the importance of reaction time as a variable, which often can mask the effects of other important reaction parameters when evaluated as an independent variable. For that reason, reaction times of 1, 2, and 4 h were evaluated for each experimental condition described in Figure 1, Figure 2 and Figure 3. Central points are the average of levels +1, and +1 and axial points were determined by interpolation (α = ±1.41). Coded and uncoded levels and their corresponding independent variables are shown in Table 1. In sum, the effects of temperature and amount of enzyme were simultaneously evaluated by the use of a central composite rotatable design, totaling 11 experimental conditions for each reaction time evaluated. Dependent variables (i.e., evaluated response) were the percentage of lactose hydrolysis and the abundance of major oligosaccharides (see Section 3.4 and Section 3.5). Data were analyzed by the Protimiza Experiment Design Software (http://experimental-design.protimiza.com.br). The significance of the model was tested by Analysis of Variance (ANOVA).

### 3.4. Quantification of Lactose, Glucose, and Galactose in Goat Whey

The concentration of simple sugars lactose, glucose, and galactose was determined by high-performance anion-exchange chromatography with pulsed amperometric detection (HPAEC-PAD ICS-5000+; Thermo Scientific, Sunnyvale, CA, USA). Calibration curves (R^2^ > 0.999) were prepared for glucose, galactose, and lactose (Sigma, St., Louis, MO, USA).

Goat whey permeate samples were diluted 10 to 1000 times and filtered through a 0.2 μm membrane (Acrodisc 13 mm PES, Pall Life Sciences, Port Washington, NY, USA). For monosaccharide analysis, a 25 μL aliquot was injected into a Carbo-Pac PA10 (4 × 250 mm) with a CarboPac PA 10 guard-column (4 × 50 mm) and detected by a disposable gold working electrode and a quadruple potential waveform (Dionex, Sunnyvale, CA, USA) at 1.2 mL/min flow rate and room temperature, with an isocratic condition of 10 mM NaOH for 12 min and a gradient up to 100 mM for the next 12.5 min.

### 3.5. Profile and Relative Quantification of Major Goat Whey Oligosaccharides

Two volumes of −30 °C pure ethanol were added to the whey permeate samples (1 mL), and the remaining proteins were precipitated at −30 °C for 60 min. The samples were then centrifuged at 4000× *g* for 30 min at 4 °C to recover the supernatant and dried in a vacuum centrifuge at 37 °C. Once dried, samples were resuspended in one volume of nano-pure water. CMO isolation was achieved by using a 96-well plate with porous graphitized carbon solid-phase extraction columns (PGC−SPE, 40 μL media bed volume, 2000 μg binding capacity, Glygen Corp., Columbia, MD, USA). Samples were loaded into the plate wells, where any remaining salts, monosaccharides, and disaccharides were removed by washing six times with 200 μL of water and spinning at room temperature for 3 min at 277 g. The CMO bounded to PGC column were eluted with 600 μL of 40% acetonitrile with 1% trifluoracetic acid in water (*v/v*). Purified oligosaccharides were dried by speed vacuum centrifugation at 37 °C. Samples were reconstituted in water to achieve a 200-fold dilution. To minimize instrument variation effect, samples were spiked with an oligosaccharide not present in this set of samples, 3′ sialyllactosamine (3′SLN), achieving a concentration of 1 μL/mL.

Major CMO abundances and types were analyzed using a nano-LC Quadruple Time of Flight Mass Spectrometer (Agilent 6520 accurate-mass nano-LC Q-ToF MS, Agilent Technologies, Santa Clara, CA, USA) with a microfluidic nano-electrospray graphitized carbon column chip. The analysis was carried out using a previously published method [21] with some modifications. Modifications included a data acquisition in the range 300–2500 mass/charge (*m/z*), and electrospray capillary voltage 1400–2000 V. Data analysis was performed on Agilent Mass Hunter Quantitative Analysis and Agilent Mass Hunter Profinder, version B.06.00 using the “find by molecular feature” algorithm. Extracted ion chromatograms were built for each known oligosaccharide *m/z* with a 20 ppm mass tolerance. Using an in-house built library of CMO, oligosaccharides were identified by mass and retention time. Once standardized by the peak area of 3′SLN, peak areas of the all major identified CMO were used to compare their relative abundances at the various experimental conditions.

## Figures and Tables

**Figure 1 molecules-24-03294-f001:**
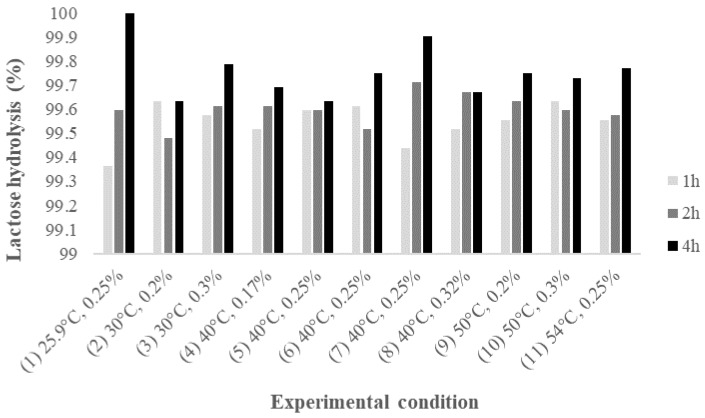
Effects of temperature and amount of enzyme on lactose hydrolysis (%) (1, 2, and 4 h) for each experimental condition evaluated as determined by HPAEC-PAD.

**Figure 2 molecules-24-03294-f002:**
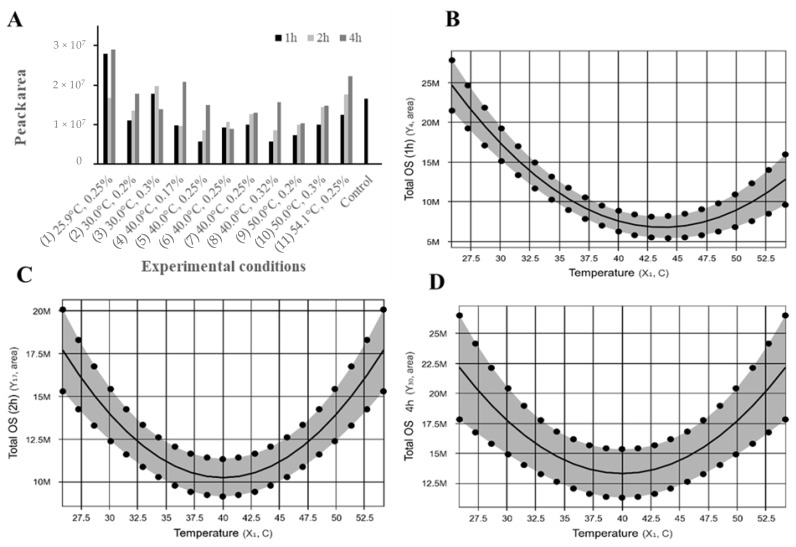
(**A**) Sum of peak areas of all oligosaccharides extracted from ion chromatograms from nano-LC Q-ToF MS analysis of total oligosaccharides on control whey and whey samples after treated for 1, 2, and 4 h according to the experimental conditions described in Table 1. Contour curve for total oligosaccharides abundance at 1 h (**B**), 2 h (**C**), and 4 h (**D**) of reaction calculated from peak areas of extracted ion chromatograms from nano-LC Q-ToF MS analysis.

**Figure 3 molecules-24-03294-f003:**
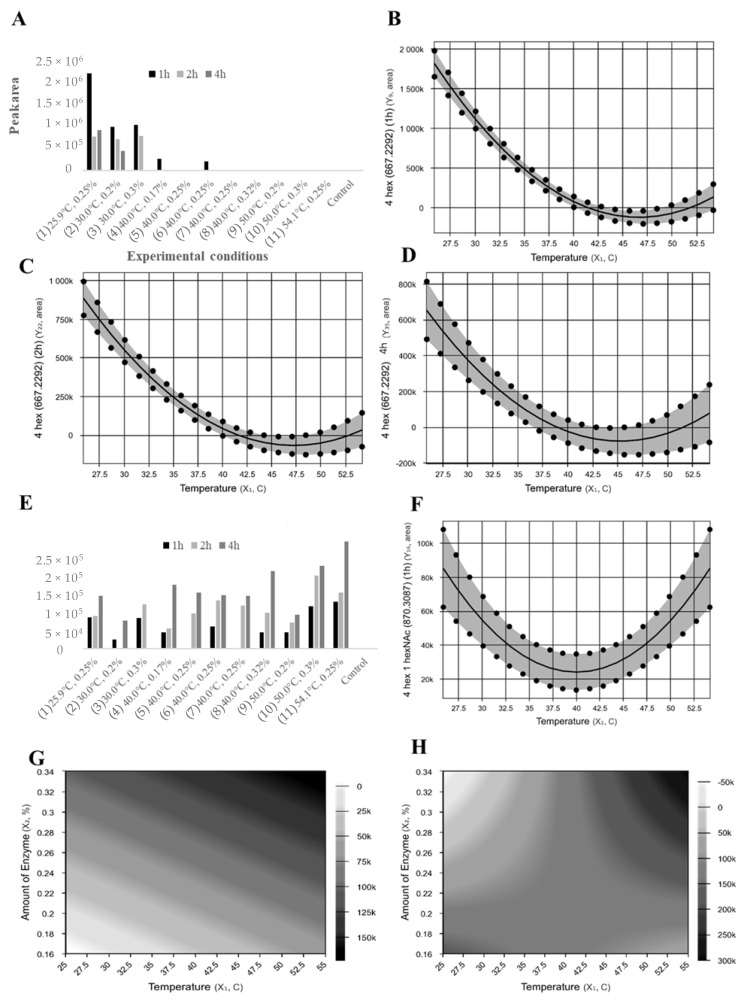
Peak areas of extracted ion chromatograms from nano-LC Q-ToF MS analysis and contour curve for the abundance of the formed neutral oligosaccharides. (**A**–**D**), 4 Hexose (Hex) (667.2292) (1 h, 2 h, and 4 h, respectively); (**E**–**H**), 4 Hex 1 Hex with either N-Acetyl-Glucosamine or N-Acetyl-Galactosamine (NAc) (870.3087) (1 h, 2 h, and 4 h, respectively).

**Table 1 molecules-24-03294-t001:** Changes in the total abundance^1^ of neutral and acidic oligosaccharides expressed as percentage (increase or decrease) of oligosaccharides in samples after treatments at 1, 2, 4 h compared to the control.

Experimental Conditions	Neutral (%)			Acidic (%)			Total (%)		
	1 h	2 h	4 h	1 h	2 h	4 h	1 h	2 h	4 h
(1) 25.9 °C, 0.25%	92	−45	−5	40	57	173	69	1	75
(2) 30.0 °C, 0.2%	−49	−39	−60	−13	8	91	−33	−18	7
(3) 30.0 °C, 0.3%	−14	−27	−83	34	77	68	8	19	−16
(4) 40.0 °C, 0.17%	−72	−92	−87	−1	21	166	−40	−42	26
(5) 40.0 °C, 0.25%	−92	−93	−88	−34	7	87	−66	−48	−10
(6) 40.0 °C, 0.25%	−81	−89	−93	2	31	13	−44	−35	−46
(7) 40.0 °C, 0.25%	−78	−89	−99	8	57	75	−39	−24	−21
(8) 40.0 °C, 0.32%	−95	−94	−92	−28	8	101	−65	−48	−6
(9) 50.0 °C, 0.2%	−92	−99	−95	−11	35	34	−56	−39	−38
(10) 50.0 °C, 0.3%	−93	−94	−86	27	88	84	−39	−13	−10
(11) 54.1 °C, 0.25%	−92	−91	−90	58	127	189	−25	6	34

^1^ Abundances are calculated based on peak areas of extracted ion chromatograms (sum of all isomers for each oligosaccharide) from nano-LC Q-ToF MS analysis of oligosaccharides on control whey and whey samples after treatment for 1, 2, and 4 h from 11 experimental conditions.

**Table 2 molecules-24-03294-t002:** Changes in the abundance^1^ of specific acidic oligosaccharides expressed as percentage increase or decrease of oligosaccharides in the samples after treatments at 1, 2, 4 h compared to the control.

Experimental Conditions	2 Hex 1 NeuAc (634.2189)			2 Hex 1 NeuGc (650.2139)			1 Hex 1 HexNAc 1 NeuAc (675.2455)			3 Hex 1 NeuAc (796.2718)			2 Hex 1 HexNAc 1 NeuAc (837.2983)			3 Hex 1 NeuGc (812.2667)		
	1 h	2 h	4 h	1 h	2 h	4 h	1 h	2 h	4 h	1 h	2 h	4 h	1 h	2 h	4 h	1 h	2 h	4 h
(1) 25.9 °C, 0.25%	43	96	213	58	101	202	−16	−100	53	54	−100	−100	65	−100	105	−100	−100	−100
(2) 30.0 °C, 0.2%	1	29	118	6	32	126	−100	−100	29	−100	−100	−100	−33	−4	−100	33	−100	−100
(3) 30.0 °C, 0.3%	58	116	95	60	109	97	−100	−100	−10	−100	−100	−100	32	67	−100	3	−100	47
(4) 40.0 °C, 0.17%	26	50	220	21	39	213	−100	−100	19	−100	−100	−100	−100	−2	−100	−37	−100	−100
(5) 40.0 °C, 0.25%	−17	35	122	−16	31	120	−100	−100	−3	−100	−100	−100	−100	−100	−100	−100	−35	−100
(6) 40.0 °C, 0.25%	26	52	38	27	58	50	−100	−25	−100	−100	−100	−100	−100	−100	−100	12	−100	−100
(7) 40.0 °C, 0.25%	22	82	102	23	81	109	−33	13	0	−100	−100	−100	−26	−100	−100	−100	−100	−23
(8) 40.0 °C, 0.32%	−20	38	141	−21	32	131	−28	−100	3	−100	−100	−100	−100	−100	−100	−100	−35	−100
(9) 50.0 °C, 0.2%	13	67	68	10	58	68	−100	−100	−100	−100	−100	−100	−100	8	−100	−100	−100	−100
(10) 50.0 °C, 0.3%	64	141	121	53	131	118	−100	−100	−15	−100	−100	−100	−100	−100	−100	−100	−100	−100
(11) 54.1 °C, 0.25%	87	202	274	71	160	249	−25	−100	−100	−100	−100	−100	12	−100	−100	−100	−100	−100

^1^ Abundances are calculated based on peak areas of extracted ion chromatograms (sum of all isomers for each oligosaccharide) from nano-LC Q-ToF MS analysis of oligosaccharides on control whey and whey samples after treatment for 1, 2, and 4 h from 11 experimental conditions.

**Table 3 molecules-24-03294-t003:** Changes in the abundance^1^ of specific neutral oligosaccharides expressed as percentage increase or decrease of oligosaccharides in the samples after treatments at 1, 2, 4 h compared to the control.

Experimental Conditions	3 Hex (505.1764)			2 Hex 1 HexNAc (546.2029)			3 Hex 1 HexNAc (708.2558)			2 Hex 2 HexNAc (749.2823)		
	1 h	2 h	4 h	1 h	2 h	4 h	1 h	2 h	4 h	1 h	2 h	4 h
(1) 25.9 °C, 0.25%	43	−17	−31	−83	−100	−100	209	−50	34	−10	−1	81
(2) 30.0 °C, 0.2%	−36	−44	−51	−100	−100	−100	−42	−10	−70	−59	−33	34
(3) 30.0 °C, 0.3%	−31	−45	−88	−100	−100	−100	30	6	−100	−11	−1	71
(4) 40.0 °C, 0.17%	−59	−88	−100	−100	−100	−100	−69	−100	−100	−62	−44	40
(5) 40.0 °C, 0.25%	−81	−94	−100	−100	−100	−90	−100	−100	−100	−49	−40	−2
(6) 40.0 °C, 0.25%	−79	−91	−100	−100	−100	−100	−83	−100	−100	−43	−3	−35
(7) 40.0 °C, 0.25%	−80	−91	−100	−100	−100	−100	−68	−100	−100	−47	−5	−100
(8) 40.0 °C, 0.32%	−93	−100	−100	−100	−100	−92	−100	−100	−90	−63	−40	−100
(9) 50.0 °C, 0.2%	−86	−100	−100	−100	−100	−100	−100	−100	−90	−44	−100	−100
(10) 50.0 °C, 0.3%	−95	−100	−100	−100	−100	−81	−100	−90	−85	−38	−100	−100
(11) 54.1 °C, 0.25%	−94	−100	−100	−100	−100	−100	−100	−100	−82	−34	−5	−100

^1^ Abundances are calculated based on peak areas of extracted ion chromatograms (sum of all isomers for each oligosaccharide) from nano-LC Q-ToF MS analysis of oligosaccharides on control whey and whey samples after treatment for 1, 2, and 4 h from 11 experimental conditions.

**Table 4 molecules-24-03294-t004:** Regression models, coefficient of determination (R^2^) of the estimated regression models for the variation in the relative abundances of total and individual oligosaccharides.

Oligosaccharides	Reaction Time (h)	Estimated Regression Model (X_1_ and X_2_)*	R^2^	F_cal_	F_tab_
Total OS	1	Y = 7507678.98 − 4194600.74X_1_ + 5592810.56X_1_²	82	17.8	4.46
	2	Y = 10222162.09 + 3729337.17X_1_²	58	12.4	5.12
	4	Y = 13284143.04 + 4422143.16X_1_²	37	5.2	5.12
3 Hex (505.1764)	1	Y = 389925.19 − 782063.92X_1_ + 502451.91X_1_²	91	38.3	4.46
	2	Y = 162313 − 580744.02X_1_ + 361604.84 X_1_²	99	293.3	4.46
	4	Y = −9339.04 − 405112.37X_1_ + 345042.31X_1_²	85	22.8	4.46
2 Hex 1 HexNAc (546.2029)	1	Y = −26531.65 − 79733.01X_1_ + 92860.76X_1_²	56	5.12	4.46
	4	Y = 90116 + 103487.67X₂ + 129100.25X_1_X₂	52	4.4	3.11^+^
2 Hex 1 NeuAc (634.2189)	1	Y = 3953983.94 + 1058216.71X_1_²	45	7.4	5.12
	2	Y = 5495767.29 + 978961.02X_1_ + 1727382.22X_1_²	68	8.5	4.46
2 Hex 1 NeuGc (650.2139)	1	Y = 2291139.71 + 614660.53X_1_²	52	9.6	5.12
	2	Y = 3167139 + 866256.25X_1_²;	50	8.9	5.12
4 Hex (667.2292)	1	Y = 69029.23 − 594727.41X_1_ + 449716.12X_1_²	95	81.5	4.46
	2	Y = 40340.74 − 300599.11X_1_ + 208263.86X_1_²	92	43.7	4.46
	4	Y = −26116.87 − 203058.25X_1_ + 194607.65X_1_²	74	11.6	4.46
1 Hex 1 HexNAc 1 NeuAc (675.2455)	2	Y = 553569.46 − 276784.73X_1_² − 276784.73X₂²	56	5.2	4.46
	4	Y = 702795.01 − 384992.53X_1_;	51	9.4	5.12
3 Hex 1 HexNAc (708.2558)	1	Y = 261151.22 − 2900998.59X_1_ + 2320426.11X_1_²	83	19.6	4.46
	2	Y = 863638.95 − 1191641.43X_1_	52	9.6	5.12
	4	Y = −148493.60 − 787036.48X_1_ + 1216793.56X_1_²	66	7.8	4.46
2 Hex 2 HexNAc (749.2823)	1	Y = 357971.42 + 117458.93X_1_²	44	7.2	5.12
	4	Y = 546409.06 − 534343.98X_1_	70	20.3	5.12
3 Hex 1 NeuAc (796.2718)	1	Y = −17284.95 − 51944.80X_1_ + 60497.33X_1_²	56	5.2	4.46
2 Hex 1 HexNAc 1 NeuAc (837.2983)	1	Y = 30908.84 − 125782.48X_1_ + 210815.71X_1_²	74	11.7	4.46
	4	Y = −44484.83 − 133685.98X_1_ + 155696.90X_1_²	56	5.2	4.46
4 Hex 1 HexNAc (870.3087)	1	Y = 23873.33 + 30604X_1_²	50	9.1	5.12
	2	Y = 82307.54 + 23892.27X_1_ + 30933.57X₂	70	9.3	4.46
	4	Y = 120775.07 + 45106.01X_1_ + 41941.77X_1_X₂	60	5.9	4.46

* Estimated regression models include regression coefficients statistically significant at *p* < 0.05, except the regression model for 2 Hex 1 HexNAc (546.2029) at 4 h. ^+^ which was statically significant at *p* < 0.1. Variables X_1_ and X_2_ correspond to temperature and amount of enzyme, respectively.

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
