# Peer review of "Understanding the Effects of Lactose Hydrolysis Modeling on the Main Oligosaccharides in Goat Milk Whey Permeate"

_molecules, 2019, doi:10.3390/molecules24183294_

Round 1

Reviewer 1 Report

Interesting work that will further advance your area of the research.

Strengths: Well-planned and executed experiments based on the similar
research from before. Important parameters were measured and observed
and sound conclusions on that have been made. Weakness: Introductory section could be improved in the way that gives
wider picture what this area of research is about, why is needed to do
this kind of experiments and then go to the details described in
introduction.

Author Response

Weakness: Introductory section could be improved in the way that gives
wider picture what this area of research is about, why is needed to do 
this kind of experiments and then go to the details described in 
introduction. 

Response:

The introduction section was updated (in red) to include previous studies reporting caprine milk oligosaccharides purification, justifying the need of the current study.

Reviewer 2 Report

Manuscript title:

Understanding the effects of lactose hydrolysis modelling on the main oligosaccharides in goat milk whey permeate

GENERAL COMENTS:

The manuscript presents very interesting research about possibilities of lactose hydrolysis to obtain oligosaccharides in goat milk. This research is valuable because it shows profile and relative quantification of major goat whey oligosaccharides obtained.

 Introduction and discussion section should be improved and other similar research should be mentioned as proposed.  More information about research done regarding lactose hydrolysis should be in introduction section. After that Authors should explain in detail what new aspects they propose in their study.

INTRODUCTION SECTION

Introduction section is rather well organized but it should be completed and mentioned other results about obtaining oligosaccharides by lactose hydrolysis – proposed references are below.

RESULT AND DISCUSSION SECTION

In result section authors should add statistical results of experiment – in column figures it should be described by letters specifying which means differ from each other. Please try to discuss in more depth other results of lactose hydrolysis. For example Neri et al. (2009) proposed the synthesis of galacto-oligosaccharides (GOS) by the action of Aspergillus oryzae β-galactosidase free and immobilized on magnetic polysiloxane-polyvinyl alcohol. In this work authors declare that GOS formation was not considerably affected by pH and temperature – please discus and compare this result. Other authors applied the response surface methodology (RSM) in terms of individual experimental factors effect estimation, their mutual interaction identification and finally, the determination of optimum conditions for highest GOS yield achievement. They also observed that the temperature and pH have no significant impact on the GOS yield, please discuss this results.

MATERIALS AND METHODS

Goat milk source and whey production

Please add information how pH was measured – kind and type of equipment is necessary. Add method used for whey lactose concentration measurement.  

Carevic, M., Banjanac, K., Milivojevic, A., Corovic, M., & Bezbradica, D. (2017). Optimization of galacto-oligosacharides synthesis using response surface methodology. Food and Feed Research, 44(1), 1–10. https://doi.org/10.5937/ffr1701001c

Dutra Rosolen, M., Gennari, A., Volpato, G., & Volken De Souza, C. F. (2015). Lactose Hydrolysis in Milk and Dairy Whey Using Microbial β-Galactosidases. Enzyme Research, 2015. https://doi.org/10.1155/2015/806240

Neri, D. F. M., Balcão, V. M., Costa, R. S., Rocha, I. C. A. P., Ferreira, E. M. F. C., Torres, D. P. M., … Teixeira, J. A. (2009). Galacto-oligosaccharides production during lactose hydrolysis by free Aspergillus oryzae β-galactosidase and immobilized on magnetic polysiloxane-polyvinyl alcohol. Food Chemistry, 115(1), 92–99. https://doi.org/10.1016/j.foodchem.2008.11.068.

Oliveira, D. L., Wilbey, R. A., Grandison, A. S., Duarte, L. C., & Roseiro, L. B. (2012). Separation of oligosaccharides from caprine milk whey, prior to prebiotic evaluation. International Dairy Journal, 24(2), 102–106. https://doi.org/10.1016/j.idairyj.2011.12.012

Rodriguez-Colinas, B., Fernandez-Arrojo, L., Ballesteros, A. O., & Plou, F. J. (2014). Galactooligosaccharides formation during enzymatic hydrolysis of lactose: Towards a prebiotic-enriched milk. Food Chemistry, 145, 388–394. https://doi.org/10.1016/j.foodchem.2013.08.060

Torres, D. P. M., Gonçalves, M. do P. F., Teixeira, J. A., & Rodrigues, L. R. (2010). Galacto-Oligosaccharides: Production, properties, applications, and significance as prebiotics. Comprehensive Reviews in Food Science and Food Safety, 9(5), 438–454. https://doi.org/10.1111/j.1541-4337.2010.00119.x

Thum, C., Cookson, A., McNabb, W. C., Roy, N. C., & Otter, D. (2015). Composition and enrichment of caprine milk oligosaccharides from New Zealand Saanen goat cheese whey. Journal of Food Composition and Analysis, 42, 30–37. https://doi.org/10.1016/j.jfca.2015.01.022

Author Response

INTRODUCTION SECTION

Introduction section is rather well organized but it should be completed and mentioned other results about obtaining oligosaccharides by lactose hydrolysis – proposed references are below.

Response:

Thank you for your references. More information was added to the introduction to justify current study (in red).

RESULT AND DISCUSSION SECTION

In result section authors should add statistical results of experiment – in column figures it should be described by letters specifying which means differ from each other. Please try to discuss in more depth other results of lactose hydrolysis. For example Neri et al. (2009) proposed the synthesis of galacto-oligosaccharides (GOS) by the action of Aspergillus oryzae β-galactosidase free and immobilized on magnetic polysiloxane-polyvinyl alcohol. In this work authors declare that GOS formation was not considerably affected by pH and temperature – please discus and compare this result. Other authors applied the response surface methodology (RSM) in terms of individual experimental factors effect estimation, their mutual interaction identification and finally, the determination of optimum conditions for highest GOS yield achievement. They also observed that the temperature and pH have no significant impact on the GOS yield, please discuss this results.

Response:

This study was designed using response surface methodology, which decreases the number of treatments and use the central point to quantify variability. This design does not allow statistical comparisons between the treatments so I cannot add letters to the column figures.

Thank you for the references. I have included some in the introduction and discussion. However, only one study analysed the effects of Aspergillus galactosidase in whey lactose (in red). The other studies focused on the maximization of GOS production from lactose solution (up to 50% w/v). The concentration of lactose highly impacts enzyme kinetics, which difficult comparisons. 

MATERIALS AND METHODS

Goat milk source and whey production

Please add information how pH was measured – kind and type of equipment is necessary. Add method used for whey lactose concentration measurement.

Response: Information added (in red). Lactose concentration method was already described in section 3.4.